# Fungal Whole-Genome Sequencing for Species Identification: From Test Development to Clinical Utilization

**DOI:** 10.3390/jof9020183

**Published:** 2023-01-29

**Authors:** Zackary Salem-Bango, Travis K Price, June L Chan, Sukantha Chandrasekaran, Omai B Garner, Shangxin Yang

**Affiliations:** Department of Pathology and Laboratory Medicine, David Geffen School of Medicine, University of California Los Angeles, Los Angeles, CA 90095, USA

**Keywords:** fungal species identification, next-generation sequencing, whole-genome sequencing, fungal infections, ITS, 28S rRNA, beta-tubulin, *k-mer* tree

## Abstract

Using next-generation sequencing (NGS), we developed and validated a whole-genome sequencing (WGS)-based clinical test for fungal species identification on clinical isolates. The identification is mainly based on the fungal ribosomal internal transcribed spacer (ITS) region as the primary marker, and additional marker and genomic analysis applied for species within the Mucorales family (using the 28S rRNA gene) and *Aspergillus* genus (using the beta-tubulin gene and *k-mer* tree-based phylogenetic clustering). The validation study involving 74 unique fungal isolates (22 yeasts, 51 molds, and 1 mushroom-forming fungus) showed high accuracy, with 100% (74/74) concordance on the genus-level identifications and 89.2% (66/74) concordance on the species level. The 8 discrepant results were due to either the limitation of conventional morphology-based methodology or taxonomic changes. After one year of implementation in our clinical laboratory, this fungal NGS test was utilized in 29 cases; the majority of them were transplant and cancer patients. We demonstrated the utility of this test by detailing five case studies, in which accurate fungal species identification led to correct diagnosis, treatment adjustment or was ruled out for hospital acquired infection. This study provides a model for validation and implementation of WGS for fungal identification in a complex health system that serves a large immunocompromised patient population.

## 1. Introduction

Since its inception, it has become clear that next-generation sequencing (NGS)-based methodologies offer a plethora of solutions to modern-day clinical and diagnostic problems. Microbial whole genome sequencing (WGS) in particular can be utilized for a wide variety of applications, including speciation and phylogenetic analysis of microorganisms, bacterial, fungal and viral strain typing, antimicrobial resistance (AMR) prediction and analysis, and genomic epidemiologic investigation and monitoring [1,2,3]. Despite years of data indicating the incredible value of NGS-based technologies, particularly in light of the recent COVID-19 pandemic [4], relatively few clinical microbiology laboratories have implemented it into their routine workflows [5].

This discrepancy is likely due to implementation of the assay being complex, labor and training intensive, and expensive and time consuming. Laboratories must also develop new quality control metrics, reference standards, and general testing infrastructure with little adaptability between respective diagnostic assays [6,7]. Finally, national guidelines for NGS in diagnostic laboratories are not nearly as robust as in other testing modalities and are extremely limited for clinical microbiology [6]. Overall, implementation of NGS in the clinical laboratory boasts a unique set of challenges but also provides a broad level of sensitivity and specificity not typically found with other testing modalities. 

For the clinical microbiology laboratory, species-specific fungal identification has always presented a difficult challenge [8]. Conventional methods (e.g., morphologic identification, biochemical tests, MALDI-TOF) may fail to offer appropriate identification at the genus or species level, or provide discrepant results, making it difficult for the clinician to tailor treatment to best benefit the patient [9,10,11,12]. WGS provides a comprehensive set of sequence data, thereby allowing for a broad analysis of highly conserved genes/regions and eventual phylogenetic identification and genotypic typing of fungal microorganisms. Such identification can provide actionable information for the clinician to appropriately treat fungal infections and prevent catastrophic consequences of fungal misidentification [13]. 

An NGS-based assay using WGS of pure fungal isolates for fungal identification was developed and implemented at the University of California Los Angeles (UCLA) Clinical Microbiology Laboratory. Herein are the wet laboratory procedures, bioinformatic analyses, and various assay quality metrics used. Validation studies to measure sensitivity, specificity, and concordance with other testing modalities were also performed. Five clinical cases where the WGS assay was implemented are described to discuss how this testing modality influenced clinical outcomes for patients. 

## 2. Materials and Methods

### 2.1. Fungal Isolates and Reference Identification Methods

A total of 75 unique fungal isolates obtained from clinical specimens or reference collections were used, including 22 yeasts, 52 molds, and 1 mushroom-forming fungus. Clinical isolates (46/75) collected from 2016–2020 and stored at −70 °C were analyzed retrospectively. Reference isolates (29/75) were obtained from the ATCC (Manassas, VA, USA) or CDC Antimicrobial Resistance (AR) isolate bank. All isolates (100%, 75/75) had genus-level IDs and most (79%, 59/75) had species-level IDs. A variety of identification methods were used, including microscopic morphology, VITEK^®^ MS (bioMérieux, Hazelwood, MO, USA), API^®^ (bioMérieux), and AccuProbe (Hologic, San Diego, CA, USA). Additionally, 6 isolates were sent to the UT Health San Antonio Fungus Testing Laboratory for identification by DNA sequence analysis. Additional information regarding source, identification method(s) used, and both the original and WGS taxonomic ID are found in Appendix A.

### 2.2. Fungal Isolate Preparation and DNA Extraction

All fungal isolates were treated with 70% ethanol and heat inactivated (100°C for 30 min). A bead beating step was added to assist with mechanical lysis of the fungal cell wall. A viability study showed that these steps resulted in no growth of a variety of diverse clinically relevant fungi, including both yeasts and molds (Appendix A). For all samples, the Qiagen (Valencia, CA, USA) EZ1 Blood and Tissue Kit as well as the EZ1 Advanced XL instrument were used (following manufacturer’s instructions) to extract genomic DNA from fungal isolates. Post extraction, DNA was quantified using Qubit 1× double-stranded DNA HS assay with the Qubit 3.0 Fluorometer (Thermo Fisher Scientific, Waltham, MA, USA). DNA quantities greater than 0.04 ng/µL were considered acceptable. PCR grade water was used as a negative extraction and sequencing control. A reference strain of *Candida tropicalis* (CDC AR Bank 3045) was used as a sequencing positive control for each run.

### 2.3. Library Preparation and Sequencing

Methods of library preparation using the Illumina DNA Prep Kit with NGS via the Illumina MiSeq (Illumina, San Diego, CA, USA), which generated 250 bp paired-end reads, has been described previously [7].

### 2.4. Bioinformatic Analysis: De novo Assembly and In-House Database Query

An overview of the analytical workflow is shown in Figure 1 and Figure 2. Data were uploaded to the Illumina BaseSpace cloud and de-multiplexed. Various sequence-run quality control (QC) metrics were recorded: percent passing filter (% PF), evenness of reads PF (i.e., ≥2% PF reads per sample), % undetermined reads, lane density, and %Q30. Results and passing criteria are listed in Appendix A. Some metrics (i.e., % PF and % Q30) were considered mandatory, meaning the sequencing run would be rejected. Other metrics were considered optimal, meaning the NGS board would decide how to proceed based on further review of the data. At least 1 million reads per sample are required for downstream bioinformatic analysis.

Sequences were uploaded to the CLC Genomics Workbench version 12 software (Qiagen), where they were trimmed and paired. A de novo sequence assembly was performed; minimum contig length was set to 1000. Various assembly QC metrics were recorded: total reads, number of contigs, N50, and % GC content. Results are listed in Appendix A.

Contigs with a range of 5000–10,000 nucleotides were randomly chosen and queried using the NCBI nucleotide BLAST, until a consensus match based on at least three queries was reached. A reference ribosomal internal transcribed spacer (ITS) region sequence of the consensus result was then chosen and downloaded from RefSeq (https://www.ncbi.nlm.nih.gov/bioproject/177353). If the ITS region sequence was not available, a sequence from a closely related species was chosen instead.

### 2.5. Bioinformatic Analysis: ITS Region Mapping and Consensus Sequence Query

Sequences were then mapped to the reference ITS region using the workflow shown in Appendix A. The average coverage and the %5X and %10X coverage were recorded. If the %5× and %10× coverage were both less than 80%, the mapping was repeated using the top result obtained from the database query of the consensus sequence.

The consensus ITS sequences were queried using the Westerdijk Fungal Biodiversity Institute database pairwise alignment tool (https://wi.knaw.nl/page/Pairwise_alignment) set to default settings. The reference description, percent overlap and percent similarity of the top result(s) were documented. A percent similarity cutoff of 98.5% was applied based on the international consensus for ITS sequencing for fungal species identification [14,15,16]. If the top result(s) were not part of the Mucorales family or the *Aspergillus *genus, the result was finalized using the reporting scheme in Table 1.

### 2.6. Bioinformatic Analysis: Additional Steps for Mucorales

If the top result(s) obtained from the ITS consensus sequence query of the Westerdijk database were part of the Mucorales family, an additional target gene (28S) was assessed [11]. A reference 28S sequence of the top result was downloaded from RefSeq (https://www.ncbi.nlm.nih.gov/bioproject/PRJNA51803) and the mapping protocol was repeated. The consensus 28S sequence was queried using the Westerdijk database and the results were finalized using the reporting scheme in Table 1. A percent similarity cutoff of 99.0% for the 28S sequence was applied based on our internal validation data.

### 2.7. Bioinformatic Analysis: Additional Steps for Aspergillus

If the top result(s) obtained from the ITS consensus sequence query of the Westerdijk database was part of the *Aspergillus* genus, a *k-mer* tree analysis was performed using an in-house database of 97 *Aspergillus* reference genomes (Appendix A) downloaded from the NCBI Genbank. This in-house database is updated semi-annually or as needed. Clustering of the sample with a reference genome in the same branch is indicative of the same species (Appendix A). If the sample is clustered with 2 reference genomes, then the taxonomic call will be reported as a split (e.g., *Aspergillus flavus*/*oryzae*) (Appendix A). In addition, the consensus sequence of the beta-tubulin gene was extracted and queried using both the Westerdijk and NCBI Genbank database to confirm the *k-mer* tree-based results. The minimal gene coverage for the consensus sequence is set as ≥98%; failure to meet this cutoff would warrant a repeat sequencing. The alignment thresholds are set as ≥98% overlap and ≥99% similarity; if a top hit meets these criteria, then it is accepted as the species identification. Discrepancy between the results based on the beta-tubulin gene and the *k-mer* tree analysis was mainly due to missing reference genomes in the in-house database and required a re-analysis of the *k-mer* tree by incorporating the missing reference genomes that might recently become available.

### 2.8. NGS Board

A team consisting of mainly PhD clinical microbiologists, and occasionally surgical pathologists and infectious disease specialists, reviewed the QC data, conventional microbiology results, NGS results and case history, performed literature review, and met to discuss the interpretation of the fungal identification and its clinical utility. In certain cases when there was ambiguity in taxonomic calling following the standard workflow, additional analysis was performed, including using the Genbank nt database as the secondary reference database to query both the ITS region and the 28S rRNA gene to determine the definitive identification if the top hits of both sequences were in agreement.

### 2.9. Validation

WGS-based IDs were compared to the original ID to determine concordance. The method used for the original ID was recorded and considered in subsequent discrepant analyses. Precision studies were performed. An in silico validation of the Westerdijk Fungal Biodiversity Institute was performed. Clinical utility was assessed by chart review of patients for select clinical isolates.

### 2.10. Ethics

This study was reviewed by the UCLA Human Research Protection Program and received an IRB exemption. Due to the nature of this study as a clinical test validation project, the samples were not de-identified and were handled in the same manner as the clinical samples. The results of this study were never reported to the providers or the patients. The retrospective case review was performed exclusively for this study and had no impact on patient care.

## 3. Results

### 3.1. Quality Control and Bioinformatics Performance

Twenty sequencing runs were performed from June 2019–March 2021 by seven different laboratory staff members. Several metrics were obtained from the Illumina BaseSpace Sequence Hub. QC parameters and a summary of results are provided in Appendix A. The % PF, % Q30, and acquired sequence reads per sample were deemed mandatory criteria. One isolate (UCLA_311, *Sporothrix schenckii*) was excluded from subsequent analyses due to insufficient sequence output (<1 million reads cutoff).

Sequencing metrics for each isolate were obtained from the CLC Genomics Workbench. QC parameters for de novo assembly and reference gene mapping, as well as a summary of the results, are provided in Appendix A.

Precision studies were performed between and within sequencing runs for both a yeast (*Candida tropicalis*, UCLA_374) and a mold (*Coccidioides immitis*, UCLA_293). For within-run analysis, genomic DNA from both organisms was obtained from multiple unique preparations and sequenced on the same run. For between-run analysis, the same preparation was used across three separate runs. All replicates passed all QC checks and were correctly identified to the species level at ≥99% overlap and similarity for the ITS region (Appendix A).

### 3.2. Database Validation

An in silico validation of the Westerdijk Fungal Biodiversity Institute database was performed using 22 reference sequences from a variety of yeasts and molds published in the NCBI GenBank. All (100%) had concordant IDs (Appendix A).

### 3.3. Assay Performance

Using WGS, 100% (74/74) and 93% (69/74) of isolates were identified to the genus and single-species levels, respectively. Based on our ID scheme (Figure 2), the ITS region was ultimately used for 52 IDs, 28S was used for 9 IDs (Mucorales), and both the beta-tubulin and *k-mer* tree analysis were used for 13 IDs (*Aspergillus* spp.) (Appendix A).

### 3.4. Comparison of Beta-tubulin and Calmodulin Genes vs. K-mer Tree Based Analysis for Aspergillus Species Identification

The beta-tubulin and calmodulin gene sequences were extracted and analyzed in 13 *Aspergillus* isolates. However, 10/13 isolates could not achieve sufficient gene coverage (%1× ≥ 98%) in calmodulin, which resulted in misidentifications in 4 samples and no identification in 1 sample, despite sufficient total sequence reads (>2 million) acquired in these samples, suggesting that calmodulin may probably be located in a GC-rich area that tends to be under-sequenced (Table 2). In contrast, 100% samples achieved sufficient gene coverage in beta-tubulin, which generated species identifications with 100% accordance with those determined by the *k-mer* tree-based analysis (Table 2). 

### 3.5. Concordant Results and Discordant Analysis

All (100%, 74/74) WGS IDs were concordant at the genus level; 89.2% (66/74) were concordant at the species level. The eight discordant samples are listed in Table 3. Upon further investigation, all of the discordant results can be potentially explained by the limitation of the conventional morphology-based identification methodology or recent taxonomic reclassification. For example, *A. niger* and *A. tubingensis* are often confused morphologically, with many other species within the Aspergillus section *Nigri* (black aspergilli) being misidentified for *A. niger* [17]. *Scedosporium dehoogii* is a relatively “new” species (2008) that is also morphologically similar to *S. apiospermum*. *C. albidus* was split into 12 distinct species in 2000, including *C. liquefaciens*, and is commonly misdiagnosed with conventional methods [18]. Notably, two *Aspergillus* samples showed discrepant species identifications. A reference *A. nidulans* (based on morphology) was identified by WGS as *A. quadrilineatus*, which belongs to the *Aspergillus* section Nidulantes and is both genetically and morphologically similar to *A. nidulans*. Similarly, a reference *A. ustus* was identified by WGS as *A. calidoustus* due to a recent reclassification of *A. ustus* into *Aspergilli* section Usti (group ustus), which now consists of >20 species [19].

### 3.6. Clinical Utility

The fungal WGS test was implemented in 2021 at the UCLA Clinical Microbiology Laboratory. To date, the laboratory has used the test to assist with fungal identification 29 times. A chart review was performed to assess that impact that it has had on patient outcomes. Of these 29 cases, 14% (4/29) of patients presented with cutaneous infections, 31% (9/29) with post-transplant infections, 21% (6/29) with infections likely related to treatment of malignancy, 10% (3/29) with concurrent autoimmune disease, 10% (3/29) with alcohol-related liver cirrhosis, 10% (3/29) with pulmonary disease (bronchiectasis, severe asthma, etc.), and 3% (1/29) with trauma. Overall, 72% (21/29) of these infections stemmed from patients in an immunocompromised state. Here, we present five examples of where it assisted in patient care.

Case 1: 21-year-old female with a past medical history significant for obesity, asthma, slipped capital femoral epiphysis, and three hip surgeries with a recent diagnosis of left-sided osteosarcoma with pulmonary metastases. The patient presented to the Emergency Department after total left femur endoprosthesis placement due to poor wound healing. The surgical site required irrigation and debridement, and the patient was admitted. The wound site was cultured and suspected to be yeast (*Ustilago* spp. vs. *Paecilomyces* spp.) based upon morphology, which was communicated to the clinical team taking care of the patient. When trying to speciate the isolate, both MALDI-TOF and carbohydrate assimilation tests yielded inconclusive results. The sample was deemed to be a candidate for WGS identification and was found to be *Arthrographis kalrae* using the above scheme. Additional testing (growth at 40–42 °C, weak urease positivity, growth with cycloheximide, and negativity for nitrate assimilation) confirmed this identification. The preliminary ID provided to the clinical team was amended to reflect this finding. The decision was made to amputate the patient’s leg, due to the need to continue with her second round of chemotherapy, which would present a serious challenge when considering wound source control and healing. This decision reflected the confirmed fungal ID by WGS, coupled with case reports found in the literature about this pathogen and treatment post-amputation as the patient completed her second round of chemotherapy.

Case 2: 58-year-old male with a past medical history significant for alcohol-associated liver disease, status-post orthotopic liver transplantation who presented with invasive pulmonary aspergillosis (*Aspergillus fumigatus* by morphology), a loculated pleural effusion, left-sided necrotizing pneumonia, and progressive bilateral nodular consolidations. Further studies revealed a probable diagnosis of central nervous system aspergillosis as well. He was initially started on isavuconazole, but serial radiographs showed further progression of his pulmonary disease. Therapy was modified to voriconazole, but future studies revealed voriconazole resistance by Minimum Inhibitory Concentration (MIC) testing in a background of subtherapeutic levels of systemic antimicrobials detected likely due to the patient hypermetabolizing the medication. He was switched to posaconazole, but clinicians reported difficulty achieving therapeutic levels when they consulted the UCLA Clinical Microbiology Laboratory. Three samples derived from the patient, including tracheal aspirate, lower endotracheal aspirate, and bronchoalveolar lavage, were tested using the WGS identification method. The three samples tested positive for *Aspergillus lentulus*, *Aspergillus fumigatus*, and *Aspergillus lentulus*, respectively. Identification of *A. lentulus* was clinically significant given its decreased susceptibility to all three antifungal classes and potentially high rate of mortality (>60%) [20]. Higher MICs have been observed for voriconazole/itraconazole with *A. lentulus* [21], so the decision was made to switch the patient to concurrent high-dose posaconazole and caspofungin IV therapy (300 mg IV BID, 50 mg IV q24h, respectively). Notably, co-infection of *Aspergillus fumigatus* and *Aspergillus lentulus* is not uncommon as shown in this case and others [22], highlighting the clinical necessity for accurate identification of all mold isolates despite similar morphologies.

Case 3: 47-year-old female with a past medical history significant for end stage renal disease, type two diabetes mellitus, congestive heart failure, and multiple instances of peritonitis/pneumoperitoneum who presented with abdominal discomfort. Peritoneal dialysate cultures were collected and were positive for mold. The patient was started on empiric treatment with isavuconazole due to a high clinical suspicion for *Aspergillus* spp. vs. *Coccidioides* spp. (the patient was from Bakersfield, CA, USA). Three days later, the isolate was identified as *Fusarium* sp. based upon morphological appearance. Clinicians requested WGS to speciate the sample. Upgrading treatment to voriconazole was considered, but was deemed to be high risk in this patient due to prolonged qTC and shifting electrolyte levels while on hemodialysis. WGS returned an ID of *Phaeoacremonium angustius*, a species that is often confused for *Fusarium* spp. morphologically. *Phaeoacremonium* spp. has been found to be more susceptible in vitro than *Fusarium* spp. [23]. This finding prevented unnecessary combination anti-fungal therapy and simplified the regimen for the patient.

Case 4: In this case, WGS was used to determine if there was a nosocomial outbreak of *A. niger* in an ICU. Two patients within the ICU, both status-post orthotopic liver transplantation with tracheostomies, had persistent growth of *A. niger* from multiple respiratory isolates identified via morphology. Patient 1 was a 74-year-old male with a past medical history significant for alcohol-associated liver disease/nonalcoholic steatohepatitis, refractory ascites, hepatic encephalopathy, and previous *A. niger* colonization whose aspirates were positive for both Gram-negative rods and *A. niger*. Patient 2 was a 63-year-old male with a past medical history significant for cholangiocarcinoma with biliary obstruction whose tracheal aspirate was found to contain both *A. fumigatus* and *Aspergillus niger.* Due to high suspicion of nosocomial transmission, a WGS ID was requested of both patient’s isolates. Patient 1′s isolate was identified as *A. tubingensis* whereas Patient 2′s sample was found to be *A. niger*, confirming that nosocomial transmission was not occurring and allowing for clinicians to tailor each individual’s treatment appropriately.

Case 5: 74-year-old male with a past medical history significant for hypertension, cerebral vascular accident (with subsequent cognitive deficit and fall risk), and smoking who presented to the dermatology clinic with chronic fingernail and toenail infections for the past 20 years. He also noted a history of working as a bartender, where feet and hands were constantly wet. Nail clippings were taken and showed mycotic growth somewhat consistent with *Acremonium* spp. However, subsequent MALDI-TOF failed to provide an ID, and WGS identification was initiated due to concerns of cryptic infection. WGS testing yielded an ID of *Chaetomium elatum*, a non-dermatophyte mold that has been implicated in occasional cases of onychomycoses [24]. Significantly, the treatment for *Chaetomium* spp. is unknown and very few publications discuss potential therapy options, though resistance to several antifungals has been reported in in vitro studies [25,26]. Some success with topical solutions (amorolfine vs. ciclopirox) have been reported [24]. Due to speciation results provided by WGS, topical treatment was tailored to address *Chaetomium elatum* infection, with a plan to switch to oral treatment (terbinafine) if symptoms did not improve.

## 4. Discussion

In this study, a WGS-based assay for fungal identification was developed and implemented for use in the clinical microbiology laboratory. The assay utilizes the Westerdijk Fungal Biodiversity Institute database, which was validated for use in our laboratory. Quality control metrics at the bench, during sequencing, and within bioinformatic analysis, were also developed and validated using a diverse set of clinically relevant fungal isolates. Precision and concordance were also recorded during the validation. The test was deployed in 2021 and is currently in use at the UCLA Clinical Microbiology Laboratory. To date, it has been used to help care for 29 patients, many in an immunocompromised state due to transplantation, malignancy, liver cirrhosis or autoimmune diseases.

The assay utilizes ITS as the primary gene for differentiation of fungal species, followed by 28S rDNA for the Mucorales family, and beta-tubulin and *k-mer* tree analysis for *Aspergillus* spp. In our validation set of samples, the ITS region alone was used to identify 70% (52/74) of samples, followed by 12% (9/74) for 28S rRNA (all Mucorales) and 25% (13/74) for beta-tubulin and *k-mer* tree (all *Aspergillus* spp.). Notably, although the calmodulin gene has also been endorsed as a reliable marker for *Aspergillus* speciation [11,12], in our study, we observed insufficient coverage of this gene, which prevented us from incorporating it into our analytical algorithm. We identified 100% (74/74) of the samples to the genus level, with 100% concordance when comparing to other identification methods. The assay generated single species results for 93% (69/74) of samples and multiple species (same genus) for 4% (3/74) of isolates. Species-level discordance was low (8.1%, 6/74) and was likely due to issues with the primary ID of the isolates with similar morphologies compared to the species identified by NGS.

Although Sanger sequencing is still the standard and most widely used method for definitive fungal identification and generally meets the clinical needs, it is lacking the capability for high-resolution strain typing that can be used for genomic surveillance of significant or emerging fungal pathogens such as *Candida auris*. In addition, the same data generated by WGS also enables anti-microbial resistance (AMR) prediction, a promising area that we are actively exploring for clinical applications of molecular AMR testing (e.g., *FKS* genotyping for echinocandin resistance in *Candida* spp. [27,28] and *CYP51A* genotyping for azole resistance in *Aspergillus* spp. [29]). Further, our laboratory has already been routinely performing WGS for bacterial species identification, which is a batch test and requires more samples in one run to make it more cost effective. Since the wet-lab process of fungal species identification is identical to bacterial species identification, adding fungal samples to our WGS run actually made our tests more cost effective. This is another important difference between Sanger sequencing and WGS, as the former method requires pre-determined targets to set up different assays for different applications in the wet-lab process, while the latter can accommodate a wide spectrum of sample types by using the same wet-lab process and only relies on different data analyses to generate the desired results. Nevertheless, currently the cost of WGS is still much higher than Sanger sequencing, but this may change in the future as the cost of NGS continues to decrease due to technological advancement and increasing competition in the NGS industry. Currently, however, it should be noted that WGS is still too expensive to be the first-line method for fungal identification.

MALDI-TOF is the other commonly used method for fungal identification and has shown its strengths of cost effectiveness, speed, and accuracy compared to sequencing. However, it also has several limitations, including an incomplete database, interference by growth media, and frequent technical failures due to small or mucoid colonies [30]. In our laboratory, we use MALDI-TOF as the first-line method for fungal identification, but frequently (15–20% for molds, 1–3% for yeasts) could not acquire identification. Therefore, the WGS test described in this study is a good second-line identification method for the fungal isolates on which MALDI-TOF failed to work.

When implementing a WGS test in a clinical laboratory, it is important to recognize that there is ample opportunity for error. The UCLA Clinical Microbiology Laboratory implemented an NGS board to manage this by reviewing weekly data, helping with troubleshooting, and discussing technical or analytical issues in real time. Clinicians are actively encouraged to join board meetings to amplify the clinical picture in the discussion. Additionally, constant monitoring of the NGS-based technology in comparison to other methods of identification is necessary to ensure accuracy of results. In fact, a fungal WGS assay should be considered an auxiliary test in a typical laboratory diagnostic schema, especially when other methods yield confusing or poor results.

WGS is primarily limited by the breadth and depth of the database that a given assay queries. To date, the availability of fungal genomes for analysis is quite lacking [31], especially when compared to bacterial databases. Several groups are working on addressing this issue given the lack of data [32,33]. This poses a unique challenge, since such WGS testing typically is applied towards immunocompromised patients who are infected by environmental organisms (some of which represent novel infections) that have not previously been sequenced for clinical analysis. It also highlights why continued WGS of fungi found in the environment is important, especially since typical identification methods often fail to recognize these atypical species or potentially confuse them for other pathogens. As databases expand, the sensitivity and specificity of WGS-based fungal identification should also improve. Therefore, it is of paramount importance that clinical microbiology laboratories continually assess the fidelity of their reference database to confirm its usefulness. In the case of the Westerdijk Fungal Biodiversity Institute database, our laboratory completes a biannual assessment using a set of 22 isolates from the original validation.

The challenges of implementing a WGS-based fungal ID assay can be immense, particularly for laboratories that lack a WGS bioinformatics pipeline for validation and implementation. At this time, assays of this type may not be appropriate in community treatment centers, given the generally immunocompetent patient population they serve and the availability of other fungal ID methods. However, when typical ID methods fail, community hospitals should consider either referring patients to a center with more sensitive diagnostics or sending samples to laboratories equipped to complete sequence-based assays. At centers that cater to chronically immunocompromised patients (tertiary or quaternary), integrated WGS would serve a clinical utility, particularly in situations where clinicians liaise directly with the clinical microbiology laboratory. Additionally, WGS provides a unique opportunity to evaluate potential mycotic outbreaks, especially in the nosocomial setting [34]. For example, our group recently used a modified version of this WGS assay to evaluate and characterize an outbreak of multidrug-resistant Clade III *Candida auris* in Los Angeles [34,35]. This flexibility of fungal WGS assays could prove essential given the changing dynamics of invasive fungal infections worldwide.

The global burden of invasive fungal infections is incredibly difficult to estimate due to a wide array of challenges ranging from sample collection and culture methods to ineffective diagnostic assays. However, the global burden is increasing—likely due to the growing numbers of persons living with human immunodeficiency virus, those receiving immunomodulators, increasing fungal resistance to typical treatment modalities, and the overall aging trend of the human population [36,37]. Additionally, international travel continues to grow at exponential rates and is accompanied by climate change, which is influencing the distribution of fungi and human-fungal exposure [36,38]. Diversified identification methods as well as increased laboratory capacity is essential from both a patient care perspective and an epidemiological one.

Overall, the implementation of fungal WGS in the clinical microbiology setting is a daunting yet essential task for referral centers and central laboratories. Sensitive and specific diagnostic modalities as described in this study are needed to address the growing burden of cryptic fungal infections that typical diagnostic methods fail to recognize. The international clinical microbiology community will need to surmount the lack of standards and guidelines, financial barriers, and technical and infrastructure challenges to meet this need. This study provides a model for validation and implementation of WGS for fungal identification in a complex health system that serves a large immunocompromised patient population.

## Figures and Tables

**Figure 1 jof-09-00183-f001:**
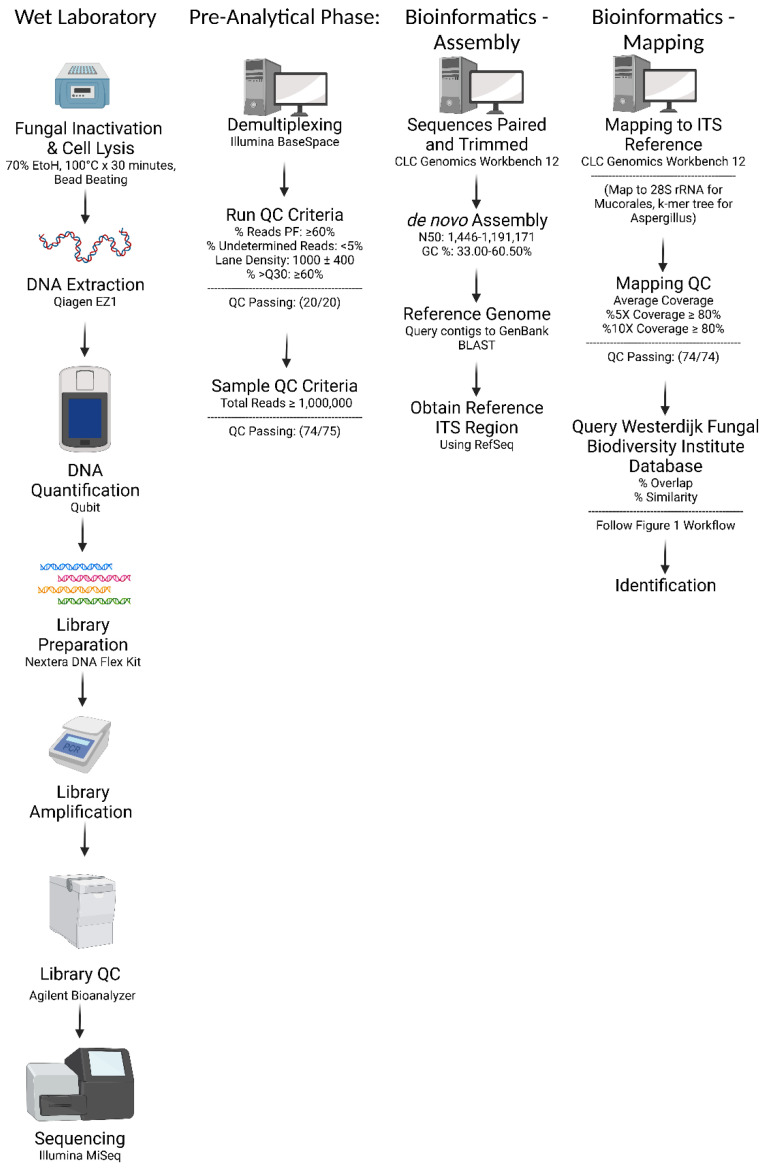
The Workflow of the WGS Test for Fungal Identification.

**Figure 2 jof-09-00183-f002:**
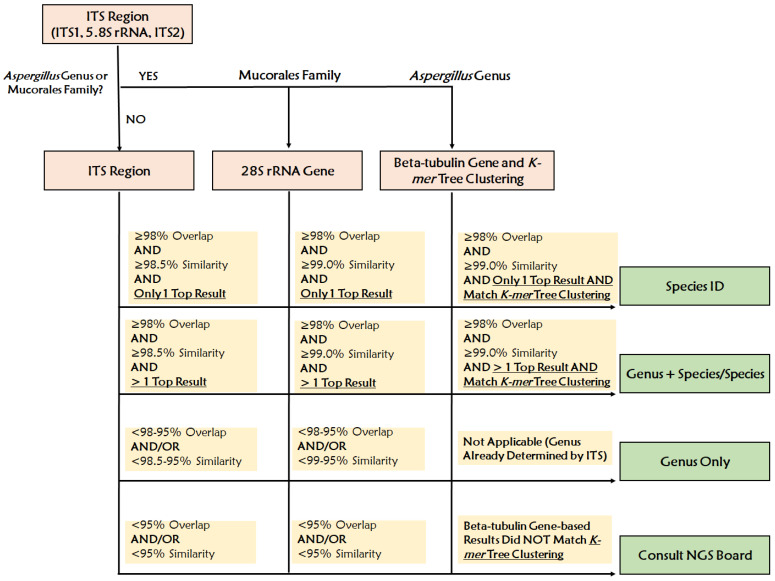
The Workflow of the WGS Test for Fungal Identification.

**Table 1 jof-09-00183-t001:** Reporting Scheme Used Following Query of the Westerdijk Fungal Biodiversity Institute Database.

ITS Query Result	28S rRNA Query Result (For Mucorales Only)	ID Chosen	Reporting Scheme	Example
≥98% Overlap AND	≥98% Overlap AND	Top Result	Genus + Species	*Candida albicans*
≥98.5% Similarity AND	≥99.0% Similarity AND			
Only 1 Top Result	Only 1 Top Result			
≥98% Overlap AND	≥98% Overlap AND	Top Results	Genus + Species/Species	*Candida albicans/dubliniensis*
≥98.5% Similarity AND	≥99.0% Similarity AND			
>1 Top Result	>1 Top Result			
<98–95% Overlap AND/OR	<98–95% Overlap AND/OR	Top Result	Genus only	*Candida* spp. most closely related to *C. albicans*
<98.5–95% Similarity	<99–95% Similarity			
<95% Overlap AND/OR	<95% Overlap AND/OR	Up to the Order Level	Consult NGS Board	Filamentous basidiomycetes, identified to be in the order of Polyporales, not able to be identified to the genus and species level
<95% Similarity	<95% Similarity			

This reporting scheme was not used for *Aspergillus* spp. Instead, the beta-tubulin gene and the *k-mer* analysis was used.

**Table 2 jof-09-00183-t002:** Aspergillus species identification based on *k-mer* Tree, beta-tubulin, and calmodulin gene analysis.

Sample ID	Total Sequence Reads	Identification Based on *K-mer* Tree	Beta-Tubulin Gene Coverage (%1X)	Top Beta-Tubulin Hit (%Overlap, %Similarity)	Calmodulin Gene Coverage (%1X)	Top Calmodulin Hit (%Overlap, %Similarity)
UCLA_160	2,522,136	*Aspergillus terreus*	99.18	*Aspergillus terreus* (100, 100)	**93.56**	***Aspergillus felis* (59, 79.57)**
UCLA_261	2,182,810	*Aspergillus tubingensis*	100	*Aspergillus tubingensis* (100, 100)	0	*N/A*
UCLA_262	2,638,830	*Aspergillus niger*	100	*Aspergillus niger (100, 100)*	**75.56**	*Aspergillus niger* (87, 99.94)
UCLA_274	5,580,086	*Aspergillus quadrilineatus*	100	*Aspergillus quadrilineatus* (100, 100)	**85.21**	*Aspergillus quadrilineatus* (100, 96.29)
UCLA_280	3,325,202	*Aspergillus terreus*	99.18	*Aspergillus terreus* (100, 100)	**93.56**	***Aspergillus felis* (59, 79.42)**
UCLA_281	3,112,042	*Aspergillus fumigatus*	100	*Aspergillus fumigatus* (100, 100)	100	*Aspergillus fumigatus* (100, 100)
UCLA_289	2,775,140	*Aspergillus fumigatus*	100	*Aspergillus fumigatus* (100, 100)	100	*Aspergillus fumigatus* (100, 100)
UCLA_297	3,402,024	*Aspergillus sydowii*	99.64	*Aspergillus sydowii* (100, 100)	**83.74**	***Aspergillus puulaauensis* (100, 94.51)**
UCLA_305	3,501,422	*Aspergillus flavus*	100	*Aspergillus flavus* (100, 100)	**75.56**	*Aspergillus flavus* (100, 99.85)
UCLA_306	2,043,130	*Aspergillus calidoustus*	100	*Aspergillus calidoustus* (100, 100)	**76.6**	*Aspergillus calidoustus* (98.13, 100)
UCLA_312	1,778,822	*Aspergillus flavus*	100	*Aspergillus flavus* (100, 100)	**75.56**	*Aspergillus flavus* (100, 99.76)
UCLA_417	2,696,552	*Aspergillus terreus*	100	*Aspergillus terreus* (100, 100)	**93.56**	***Aspergillus felis* (61, 79.42)**
UCLA_536	1,350,196	*Aspergillus fumigatus*	100	*Aspergillus fumigatus* (100, 100)	**94.93**	*Aspergillus fumigatus* (100, 100)

Insufficient coverage and incorrect identification is bolded.

**Table 3 jof-09-00183-t003:** List of Discordant Results.

Sample ID	WGS ID	Original ID	Conventional Identification Method
UCLA_24	*Scedosporium dehoogii*	*Scedosporium apiospermum*	Microscopic morphology
UCLA_144	*Scedosporium dehoogii*	*Scedosporium apiospermum*	Microscopic morphology
UCLA_156	*Candida spp.* most closely related to *C. haemulonii*	*Candida haemulonii*	Reference Isolate
UCLA_261	*Aspergillus tubingensis*	*Aspergillus niger*	Microscopic morphology
UCLA_274	*Aspergillus quadrilineatus*	*Aspergillus nidulans*	Microscopic morphology
UCLA_291	*Coprinellus* spp. most closely related to *C. micaceus*	*Coprinellus micaceus*	MALDI-TOF MS
UCLA_306	*Aspergillus calidoustus*	*Aspergillus ustus*	Reference Isolate
UCLA_415	*Cryptococcus liquefaciens*	*Cryptococcus albidus*	API 20C

## Data Availability

The data presented in this study are available on request from the corresponding author.

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
