# Peer review of "Fungal Whole-Genome Sequencing for Species Identification: From Test Development to Clinical Utilization"

_jof, 2023, doi:10.3390/jof9020183_

Round 1

Reviewer 1 Report (Previous Reviewer 3)

The authors have addressed my comments in the previous rounds of review.

Author Response

Thank you so much for reviewing this manuscript!

Reviewer 2 Report (New Reviewer)

I read with interest the manuscript entitled: " Fungal Whole-genome Sequencing for Species Identification: From Test Development to Clinical Utilization".

I was really disappointed because I found the title misleading. Indeed, although the authors used a whole genome sequencing (WGS) technology, they only analyzed the ribosomal internal transcribed spacer (ITS) region and the 28S rRNA gene. Thus their WGS approach adds nothing new compared to the conventional Sanger sequencing of these two genomic regions.

I did not understand the rational of using K-mer tree clustering of ITS sequences for Aspergillus spp. identification. It would be much more effective to analyze the beta tubulin (BTUB) region which performs better in discriminating distinct Aspergillus spp. than ITS, and should easily be extracted from the WGS reads.

Another point is that the authors insist on justifying the clinical relevance of adequate fungal identification in the clinical laboratory. I am surprised that in 2023, clinical microbiologists working in a well-known university hospital, still feel that this point needs to be justified. I was also surprised by the low number (46 from 2016 to 2020) of clinical fungal isolates for which an adequate identification has been deemed necessary in this clinical laboratory.

Author Response

Response: thank you for your comment. We have re-analyzed our data focusing on beta-tubulin gene for Aspergillus, and found that it does perform extremely well, with 100% concordance with the results determined by the k-mer tree analysis. Specifically, our initial results showed discrepancies in 2 Aspergillus isolates; upon re-analysis, it was found that our in-house Aspergillus database was missing the 2 novel reference genomes due to recent re-classification of A. nidulans and A. ustus. By including the new reference genomes, we achieve 100% agreement between beta-tubulin and the k-mer tree results. This is a great lesson for us to be mindful of the nature of constant taxonmomic changes which can have direct impact on the accuracy of fungal species identification. We really appreciate having this opportuity to improve our pipeline. Please see track changes in the text and newly added Table 3.

Response: thank you for your comment. We certainly agree with the review’s point that accurate identification of fungal isolate is absolutely necessary in the clinical setting. However, this has been challenging due to the limitation of the conventional methods for fungal identification. We started the WGS test development as early as 2018, and have already implemented this test since 2021. However, convincing the community that WGS is one of the viable solutions for solving the current fungal identification challenges is not an easy task, which is one reason why we had delayed the publication of this paper until we can collect enough data (cases) to demonstrate the clinical utility, as detailed in this paper.

Regarding the reviewer’s 2nd point, we think there might be a misunderstanding of the number of clinical isolates included in this study. In order to diversify the different kinds of fungi for the test development we tried to use as many reference fungal strains as possible to avoid the discrepancies between the conventional morphology-based identification and the WGS-based identification in the clinical isolates, which we anticipated to happen. This was more of a strategy for us to be able to meet the criteria for test validation. In reality, numerous clinical isolates in the past years in our lab couldn’t be accurately identified and we had to send these out to reference laboratories for further testing, before we could implemented our WGS test as described in this paper. Therefore, we believe that our laboratory is not the only one that has the clinical needs and can benefit from having the WGS test for in-house fungal identification. This is one reason we believe our paper can bring values to the academic and medical community.

Reviewer 3 Report (New Reviewer)

In this manuscript the authors claim that they have established a clinical test method based on whole genome sequencing (WGS) that can be used to identify fungal species in clinical isolates.

The method uses fungal internal ribosomal transcribed spacer (ITS) as the primary marker, aided by other markers, and genomic analysis for identification. The test showed high accuracy through validation of 74 unique fungal isolates. The usability of the method was demonstrated by testing with human pathogenic fungi. The method's accurate identification of fungal species contributes to proper diagnosis, and treatment adjustment. This study provides a model for health system validation and implementation of WGS for fungal identification and has implications for subsequent medical development studies. The manuscript can be considered for acceptance; however, please be sure to address the following few needs.

1. The diagrams in the article would have been better if they had been revised and embellished.

2. The figures and tables in the manuscript are very crude and not aesthetically pleasing; please correct them carefully to improve the quality of the figures and tables. Otherwise, my unacceptability may lead to the rejection of the manuscript for acceptance.

3. in lines 443, 444, 457, the species names in Latin need to be italicized.

4. There is not enough detail in the human experiment description section, please add more details.

Author Response

  1. The diagrams in the article would have been better if they had been revised and embellished.

Response: we have modified and updated Figure 2 to make it more visually acceptable.

  1. The figures and tables in the manuscript are very crude and not aesthetically pleasing; please correct them carefully to improve the quality of the figures and tables. Otherwise, my unacceptability may lead to the rejection of the manuscript for acceptance.

Response: we have modified and updated the tables to make them more visually acceptable.

  1. in lines 443, 444, 457, the species names in Latin need to be italicized.

Response: these species names are in the reference section that’s edited by the software we used, and the format will be modified and updated during the final production process if this paper is accepted. Therefore, we feel that there is no need to italicize these names at the review phase.

  1. There is not enough detail in the human experiment description section, please add more details.

Response: we have added more details in this section.

Round 2

Reviewer 2 Report (New Reviewer)

The authors adequately addressed my remarks. No further improvement seems necessary.

This manuscript is a resubmission of an earlier submission. The following is a list of the peer review reports and author responses from that submission.

Round 1

Reviewer 1 Report

This work presents a pipeline for species identification based on whole genome sequencing for fungi. The authors have written a nice and clearly described paper, but I struggle to understand the rationale for this.

WGS, by its own merit, is sufficient for species identification. Yet the method proposed here describes taking a single region (ITS), or a k-mer based approach for species identification. In the case of ITS, it is well documented that this isn't sufficient for species identification, and other genes such as beta tubulin and/or calmodulin should be used. However, why collect genome wide information, to then only extract one gene?

The k-mer approach for Aspergillus potentially has merit, but I still feel this is excessive. 

As it stands, I do not see the merit of this approach over maldi-tof for species ID or cal/beta-tub sanger sequencing. Perhaps this paper could be improved with some benchmarking on how this approach performs in relation to other methods (in terms of cost, time/efficiency). 

Author Response

Reviewer 1.

This work presents a pipeline for species identification based on whole genome sequencing for fungi. The authors have written a nice and clearly described paper, but I struggle to understand the rationale for this.

WGS, by its own merit, is sufficient for species identification. Yet the method proposed here describes taking a single region (ITS), or a k-mer based approach for species identification. In the case of ITS, it is well documented that this isn't sufficient for species identification, and other genes such as beta tubulin and/or calmodulin should be used. However, why collect genome wide information, to then only extract one gene?

The k-mer approach for Aspergillus potentially has merit, but I still feel this is excessive. 

As it stands, I do not see the merit of this approach over maldi-tof for species ID or cal/beta-tub sanger sequencing. Perhaps this paper could be improved with some benchmarking on how this approach performs in relation to other methods (in terms of cost, time/efficiency).

Response: we thank the reviewer for the comments. In Line 69 – 72, we have mentioned that “MALDI-TOF may fail to offer appropriate identification at the genus or species level, or provide discrepant results”. This is based on our first-hand experience, as MALDI-TOF often doesn’t work well for molds. We further demonstrated this by several case studies, including Case 1: Arthrographis kalrae is not in the MALDI-TOF database, and Case 5: MALDI-TOF failed to identify Chaetomium elatum (Line 292).

Regarding Sanger sequencing, we agree that it has a good utility and performance for general fungal identification, however, it is lacking the capability for strain typing and genomic surveillance as demonstrated by our other studies using WGS for Candida auris surveillance. Further, WGS enables anti-microbial resistance (AMR) prediction, a promising area that we are actively exploring for clinical applications. Last but not least, in our lab we routinely perform WGS for both bacterial and fungal isolates. It’s a batched run and more samples can make each WGS run more cost effective. Therefore, adding fungal ID to our WGS pipeline actually makes our test more cost effective. But we admit that currently the cost of WGS is still much higher than Sanger sequencing. We have added a paragraph detailing these factors to justify using WGS for fungal ID in the clinical laboratories and explain the key differences between Sanger sequencing and WGS (Line 325-341). We had also discussed challenges of impelementing NGS-based tests in clinical laboratories (Line 342-376).

Reviewer 2 Report

The manuscript jof-2045859 presents a new methodological approach, based on whole genome sequencing and bioinformatics, for the precise identification of fungal pathogens. The study is well conducted and the results are clearly presented.

The following minor suggestion and comments may contribute to enhancing this study:  

Page 6, line 143: The authors mention "an in-house database of Aspergillus genomes downloaded from the NCBI Genbank."  I would add a supplemental table listing the species name/strain used for this analysis, with their accession numbers. Since not all the genomes of Aspergillus species described to date are found in the Genbank, these can be updated. 

Page 7, line 199: I suggest replace "A. nigri (black Aspergillus) section" by "Aspergillus section Nigri (black aspergilli)"

Page 10, lines 313-314: Why was the partial sequence of the calmodulin gene (CaM) excluded from the bioinformatic analysis? CaM is another usual marker used in Aspergillus speciation. 

Page 10, lines 365-369: Antifungal resistance is currently a problem worldwide. A pipeline like it also could be applied to the search for mutations in genes associated with antifungal resistance; such as CYP51A, Erg11, Cdr1B, Atr1...

Author Response

Reviewer 2

The manuscript jof-2045859 presents a new methodological approach, based on whole genome sequencing and bioinformatics, for the precise identification of fungal pathogens. The study is well conducted and the results are clearly presented.

The following minor suggestion and comments may contribute to enhancing this study:  

Page 6, line 143: The authors mention "an in-house database of Aspergillus genomes downloaded from the NCBI Genbank."  I would add a supplemental table listing the species name/strain used for this analysis, with their accession numbers. Since not all the genomes of Aspergillus species described to date are found in the Genbank, these can be updated. 

Response: We have included the Supplemental Table 5 that details the 95 Aspergillus reference genomes.

Page 7, line 199: I suggest replace "A. nigri (black Aspergillus) section" by "Aspergillus section Nigri (black aspergilli)"

Response: We thank the reviewer for the nice comments and suggestion. We have edited the manuscript accordingly.

Page 10, lines 313-314: Why was the partial sequence of the calmodulin gene (CaM) excluded from the bioinformatic analysis? CaM is another usual marker used in Aspergillus speciation. 

Response: We thank the reviewer for the comments and questions. We did analyze both beta-tubulin and calmodulin genes, but there was a coverage issue on both genes. For beta-tubulin, 4/13 samples couldn’t meet the QC cutoff (>80% 5X), and 2/13 results were incorrect; for CaM, 10/13 samples couldn’t meet the QC cutoff and 6/13 results were incorrect. We have added the Supplemental Table 8, and mentioned this reason in Line 322.

Page 10, lines 365-369: Antifungal resistance is currently a problem worldwide. A pipeline like it also could be applied to the search for mutations in genes associated with antifungal resistance; such as CYP51A, Erg11, Cdr1B, Atr1...

Response: We have inserted the following: “…likely due to growing numbers of persons living with human immunodeficiency virus, receiving immunomodulators, fungal resistance to typical treatment modalities and overall aging trend of the human population” (Line 381). We also added the potential of using WGS for anti-fungal resistance testing as another key difference between Sanger sequencing and WGS (Line 328-331).

Reviewer 3 Report

The manuscript by Salem-Bango et al. presents a study which utilised WGS for fungal identification in a clinical microbiology laboratory. It is a nicely written article, and WGS was found to be highly accurate for speciation. 

While the authors' attempt to make use of WGS is appreciated, the reason why the authors opted to use WGS over PCR-sequencing is not known. After all, although WGS was performed for the fungal isolates in this study, they were only identified mostly based on the ITS region, some with 28S rRNA gene. What is the difference if a clinical laboratory chooses to Sanger-sequence the ITS/28S rRNA gene for identification instead? This may also be more advantageous than WGS since more specialised equipment and bioinformatic expertise are not needed. The authors may wish to elaborate on WGS is favoured instead.

The species identification criterion is >=98% sequence identity. Does this apply to both ITS and 28S rRNA gene? What is/are the reasons for using 98% as the cutoff? Would a 98.5% cut-off be more commonly used? (Irinyi L et al. Med Mycol . 2015 May;53(4):313-37; Hoang MTV et al. Front Microbiol . 2019 Jul 18;10:1647; Nilsson RH et al. Nucleic Acids Res . 2019 Jan 8;47(D1):D259-D264.; etc.) Also, what about the 28S rRNA gene? A 98% cutoff for the 28S rRNA gene may seem too low.

What genomes of Aspergillus species are included in the in-house database? Some Aspergillus genomes may not be available in the NCBI GenBank database. Would the authors also explore the MycoCosm database (https://mycocosm.jgi.doe.gov/mycocosm/home)? 

Line 154: Why did the authors opt to use the GenBank nt database which may not be so well curated but not the curated/authenticated RefSeq ITS database?

Lines 313-317: It is mentioned that the coverage for the beta-tubulin gene for Aspergillus species was not sufficient for identification purpose. CaM is another secondary marker very often used for Aspergillus identification. How was the coverage of CaM from the WGS? Did the authors try to use CaM for Aspergillus speciation as well?

Minor language editing is needed. A number of typos are spotted.

Author Response

Reviewer 3.

The manuscript by Salem-Bango et al. presents a study which utilized WGS for fungal identification in a clinical microbiology laboratory. It is a nicely written article, and WGS was found to be highly accurate for speciation. 

While the authors' attempt to make use of WGS is appreciated, the reason why the authors opted to use WGS over PCR-sequencing is not known. After all, although WGS was performed for the fungal isolates in this study, they were only identified mostly based on the ITS region, some with 28S rRNA gene. What is the difference if a clinical laboratory chooses to Sanger-sequence the ITS/28S rRNA gene for identification instead? This may also be more advantageous than WGS since more specialised equipment and bioinformatic expertise are not needed. The authors may wish to elaborate on WGS is favoured instead.

Response: We thank the reviewer for the comments. We have added explanation for why we chose WGS instead of Sanger sequencing for fungal species identification (Line 325-341).

The species identification criterion is >=98% sequence identity. Does this apply to both ITS and 28S rRNA gene? What is/are the reasons for using 98% as the cutoff? Would a 98.5% cut-off be more commonly used? (Irinyi L et al. Med Mycol . 2015 May;53(4):313-37; Hoang MTV et al. Front Microbiol . 2019 Jul 18;10:1647; Nilsson RH et al. Nucleic Acids Res . 2019 Jan 8;47(D1):D259-D264.; etc.) Also, what about the 28S rRNA gene? A 98% cutoff for the 28S rRNA gene may seem too low.

Response: Thank you for this wonderful comment. We have adjusted the aforementioned criterion to 98.5% for ITS and 99% for 28S rRNA as seen in Table 1/Figure 2. On re-analysis, these changes do not alter our results but do better reflect the literature as the reviewer mentioned.

Thank you for recommending the 3 reference papers supporting the 98.5% cutoff. We have included them in the citations.

What genomes of Aspergillus species are included in the in-house database? Some Aspergillus genomes may not be available in the NCBI GenBank database. Would the authors also explore the MycoCosm database (https://mycocosm.jgi.doe.gov/mycocosm/home)? 

Response: We have added the Supplemental Table 5 to specify all the Aspergillus species included in our in-house database. We were not aware of the MycoCosm database but will explore using it in the future studies. Thank you for this info.

Line 154: Why did the authors opt to use the GenBank nt database which may not be so well curated but not the curated/authenticated RefSeq ITS database?

Response: We chose nt dabase instead of RefSeq simply because nt database has much more reference genomes for fungus.

Lines 313-317: It is mentioned that the coverage for the beta-tubulin gene for Aspergillus species was not sufficient for identification purpose. CaM is another secondary marker very often used for Aspergillus identification. How was the coverage of CaM from the WGS? Did the authors try to use CaM for Aspergillus speciation as well?

Response: We thank the reviewer for the comments and questions. We did analyze both beta-tubulin and calmodulin genes, but there was a coverage issue on both genes. For beta-tubulin, 4/13 samples couldn’t meet the QC cutoff (>80% 5X), and 2/13 results were incorrect; for CaM, 10/13 samples couldn’t meet the QC cutoff and 6/13 results were incorrect. We have added the Supplemental Table 8, and mentioned this reason in Line 322.

Minor language editing is needed. A number of typos are spotted.

Response: We have made further proof reading and corrected typos and some grammatical errors.

Round 2

Reviewer 1 Report

With regards your point about MALDI-TOF not being sufficient for identification of molds, I have to disagree. MALDI-TOF is routinely used in many major medical centres and is accurate in identifying the species. There are many papers to confirm this (there is even a systematic review here in Journal of Fungi that points to MALDI-TOF being "rapid, cost-effective and easy-to-handle for microbial identification" for molds - Knoll et al. 2021). In comparison, ITS for molds is known to be difficult to get the same level of accuracy as MALDI-TOF. I appreciate the authors saying that the value of WGS is that you can also complete AMR-typing, but your paper is centred around using WGS to pull out the ITS to perform species identification. Reconsider re-writing this paper describing your pipeline for surveillance as a whole, instead.

Author Response

Response: Thank you for your comment. We should clarify that we never stated that whole-genome sequencing (WGS) is better than Sanger sequencing or MALDI-TOF in the paper. Instead, we compared their differences and pros and cons, and emphasized the niches of using WGS for fungal identification. We reviewed the suggested paper (Knoll et al. 2021) but this was about anti-fungal susceptibility and not about fungal identification. Instead we reviewed and cited another relevant review article (Patel et al. 2019, Journal of Fungi, A Moldy Application of MALDI: MALDI-ToF Mass Spectrometry for Fungal Identification) which detailed the strengths and limitations of MALDI-TOF. Please see added paragraph (Lines 347 – 353)

We have also emphasized that “Currently, however, it should be noted that WGS is still too expensive to be the first-line method for fungal identification.” (Lines 345-346) and “Therefore, the WGS test described in this study is a good second-line identification method for the fungal isolates on which MALDI-TOF failed to work.” (Lines 352-353).

Reviewer 3 Report

The authors have largely addressed comments from the previous round of review. Just one more comment:

Lines 155-162 (NGS Board): It is still unclear how the NGS board resolved the identification ambiguity. When the GenBank nt database was used as the secondary reference database, what sequence was used as the queries? ITS? Another locus? Or the entire genome? It is also mentioned that for non-Mucorales species, 28S rRNA gene was used as the secondary marker. In this case, what database was used for comparison? RefSeq or the Westerdijk database?

Author Response

Response: Thank you for your comment and further question. In the standard workflow, Wedterdijk was used the primary database for both ITS and 28S rRNA (highlighted in Line 150). For further analysis by the NGS board, both ITS and 28S rRNA were queried using the NCBI nt database and if the top hits of both sequences agreed then a definitive identification is achieved (Lines 167 – 170).